# Heart Rate Recovery Assessed by Cardiopulmonary Exercise Testing in Patients with Cardiovascular Disease: Relationship with Prognosis

**DOI:** 10.3390/ijerph20064678

**Published:** 2023-03-07

**Authors:** Amy Dewar, Lindsy Kass, Robert C. M. Stephens, Nicholas Tetlow, Terun Desai

**Affiliations:** 1Department of Psychology, Sport and Geography, School of Life and Medical Sciences, University of Hertfordshire, Hatfield AL10 9AB, UK; 2Division of Surgery and Interventional Science, University College London, London WC1E 6BT, UK; 3Department of Anaesthesia and Perioperative Medicine, University College London Hospitals, London NW1 2PG, UK

**Keywords:** heart rate recovery, exercise testing, heart failure, coronary artery disease, prognosis

## Abstract

Background: The use of exercise testing has expanded in recent decades and there is a wealth of information examining the prognostic significance of exercise variables, such as peak oxygen consumption or ventilatory measures whilst exercising. However, a paucity of research has investigated the use of recovery-derived parameters after exercise cessation. Heart rate recovery (HRR) has been considered a measure of the function of the autonomic nervous system and its dysfunction is associated with cardiovascular risk. Objectives: We aim to provide an overview of the literature surrounding HRR and its prognostic significance in patients with cardiovascular disease undertaking an exercise test. Data Sources: In December 2020, searches of PubMed, Scopus, and ScienceDirect were performed using key search terms and Boolean operators. Study Selection: Articles were manually screened and selected as per the inclusion criteria. Results: Nineteen articles met inclusion criteria and were reviewed. Disagreement exists in methodologies used for measuring and assessing HRR. However, HRR provides prognostic mortality information for use in clinical practice. Conclusions: HRR is a simple, non-invasive measure which independently predicts mortality in patients with heart failure and coronary artery disease; HRR should be routinely incorporated into clinical exercise testing.

## 1. Introduction

Cardiovascular disease (CVD) is the collective name used for disorders related to the heart and blood vessels; these diseases include hypertension, coronary artery disease (CAD) and heart failure (HF) [1]. CVD is the number one cause of global deaths, representing ~30% deaths globally; in the next decade, CVD is projected to cause 23.6 million deaths, mainly from heart disease and stroke [1]. HF is a common final stage of all heart diseases, representing an inability of the heart to pump effectively, and is correlated with morbidity and mortality [2].

Cardiopulmonary exercise testing (CPET) is an objective test which comprehensive assesses an individual’s respiratory, cardiovascular, muscular and metabolic responses to physiological stress [3]. CPET is a non-invasive test which is considered a safe procedure (one major adverse event per 2500 tests) [4]. The majority of people are suitable to undergo CPET. However, there are some absolute and relative contraindications to starting an exercise test (Table 1) [4].

CPET is already a well-established method in the evaluation of perioperative risk across a range of different surgeries; early work focused on intra-abdominal surgeries; today, data derived from CPET contribute widely to multidisciplinary shared decision-making processes and are used to identify the high-risk surgical patient and any unsuspected co-morbidities [3]. CPET provides clinicians with objective evidence for the risk stratification of perioperative morbidity and mortality. As such, in patients with suspected CAD, the latest American College of Cardiology and the American Heart Association guidelines recommend exercise the use of stress tests as an initial diagnostic assessment [5].

Traditionally, CPET evaluates variables such as peak oxygen consumption ($\dot{V}$O_2_ peak), $\dot{V}$O_2_ at the anaerobic threshold (AT), and the ventilation to carbon dioxide relationship ($\dot{V}$_E_/$\dot{V}$CO_2_) [4]. However, there has been a paucity of research focusing on physiological variables after the cessation of exercise, despite the wealth of information that can be gained in the recovery periods of CPET, one of which is heart rate recovery (HRR) [6]. This review will look at the recovery of heart rate after exercise cessation in populations with CVD, specifically HF and CAD.

Exercise is the stimulus which the autonomic nervous system (ANS) acts on. The stressor of exercise results in a reduced parasympathetic and increased sympathetic drive, increasing blood flow to the heart and skeletal muscles. At the end of exercise, autonomic recovery is associated with a large decrease in heart rate [7]. Thus, HRR can act as a non-invasive method to assess ANS function [7,8], with autonomic dysfunction indicated by a slower HRR [8]. Autonomic cardiovascular dysfunction has been shown to predict mortality [9]. Typically, mortality is attributed to either all-cause mortality (ACM) or cardiovascular mortality (CVM). ACM is viewed as an unbiased and objective endpoint. However, both these mortalities are used in clinical research [10]. This review aims to examine the prognostic significance of HRR in patients with CAD or HF undergoing CPET.

## 2. Materials and Methods

### 2.1. Literature Search Strategy

#### 2.1.1. Data Sources

Available papers, published between January 1999 and January 2021, were searched in PubMed, Scopus, and ScienceDirect in order to identify studies examining HRR. Search terms included “heart rate recovery” AND “heart failure” OR “coronary artery disease” OR “coronary heart disease” OR “heart disease” OR “prognosis”. The search was restricted to full-text published articles in the English language and conducted with human participants. Reference lists were reviewed to identify additional studies. Case studies, reviews, editorials, and book chapters were excluded.

#### 2.1.2. Study Selection

Studies were included in the review if: (a) HRR was measured through exercise testing; (b) the patient cohort included cardiovascular disease (CAD or HF); and (c) the study reported on measured outcomes of ACM and CVM. Studies were excluded if: (a) the population surveyed was under 18 years old; (b) cardiac rehabilitation was the primary aim; (c) the full text was not available; and/or (d) the text failed to meet the inclusion criteria. The study selection flow diagram shows the number of records found in the initial search, as well as the number of records excluded and included as part of the final review (Figure 1).

## 3. Literature Review

### 3.1. Mechanisms

#### 3.1.1. Background Physiology

The ANS is part of the nervous system which regulates biological systems outside of conscious control and enables individuals to maintain homeostasis. Two of its subsystems include the parasympathetic (predominantly cholinergic) and sympathetic (predominantly adrenergic) nervous systems. Heart rate is governed by the intrinsic activity of the sinoatrial node (SA node), which is innervated by parasympathetic (vagus) and sympathetic (thoracic) nerves [11].

The ANS regulates the heart’s response to exercise through increasing sympathetic tone, accompanied by concomitant parasympathetic withdrawal. Initially, the increase in heart rate is a result of parasympathetic withdrawal: as exercise intensity increases, sympathetic activation contributes to exercise tachycardia [12].

#### 3.1.2. Heart Rate Recovery

After exercise cessation, HRR is mediated through sympathetic withdrawal and parasympathetic reactivation [7]. The authors studied the characteristics of HRR in three groups: healthy adults, athletes, and patients with chronic HF. In all groups, the principal determinant of recovery was vagal reactivation. The difference found related to the rate of recovery being quicker in athletes (higher vagal tone) and slower in other patients [7]. As earlier research was known to have found an association of greater vagal activity and reductions in mortality, it was later proposed that this may be the mechanism by which HRR could predict mortality [13].

In more recent studies [14], it has been corroborated that HRR, mediated by vagal reactivation is a measure of parasympathetic function, with dysfunction (slower HRR) being associated with increased mortality [15] and faster HRR being associated with decreased mortality [16].

There appear to be two distinct phases of HRR: the rapid decline and the secondary period of slower reduction. Two predominant mechanisms underpin each stage: the first phase (<60 s) is predominantly modulated by parasympathetic reactivation (vagal activity) [7] and the second phase (>60 s) by both parasympathetic reactivation and sympathetic withdrawal [8,17] (Figure 2).

### 3.2. Heart Rate Recovery

The literature has a range of methods and protocols for determining HRR; differences include cut-offs or thresholds surrounding normal and abnormal HRR and measurement duration.

#### 3.2.1. Defining HRR

Most studies define HRR as the reduction in heart rate from peak exercise to one minute later, as first proposed by Cole et al. [13], and this is given as an absolute value, e.g., 12 bpm. This definition has been used across the literature, with a small amendment made by Shetler et al. [9] that defined it as maximum heart rate minus heart rate at a chosen time period after exercise cessation. This expanded the period over which HRR was measured, ranging from 30 s to 10 min [19,21]. Most studies recorded HRR at 1 min (HRR1) or HRR at 2 min (HRR2) after exercise cessation, as currently there is no standardised time-point or threshold.

Two studies used a different method to obtain HRR. One study included 2193 CAD patients who were followed on average for 10.2 years [22]. The authors aimed to define the best model of HRR. They reported HRR as an absolute variable and a mathematical model of the slope, encompassing a curve to fit the entire process of the decline in HRR. They concluded that HRR as an absolute measure was the better predictor of mortality. Hajdusek et al. [23] included a cohort of 78 HF patients. The authors calculated the slope of HRR after 150 s, which provided a decrease in heart rate per minute. They were unable to demonstrate a significant prognostic impact of HRR. This may partially be explained by differences in methodology employed when defining HRR.

#### 3.2.2. Time-Points and Thresholds of HR

The most common measurement and threshold for abnormal HRR1 is 12 bpm. Cole et al. [13] first determined this by using a log-rank Chi-square test and found HRR had prognostic significance in their study. This HRR1 threshold of 12 bpm has been validated many times across the literature, using Cox regression analyses and receiver operating characteristic (ROC) curve analyses [9,21,24]. Another common threshold for HRR1 is 18 bpm. Watanabe et al. [16] first used this threshold as it provided the highest log-rank χ2 statistic. There are limitations to this method as it overstates the strength of an association; however, in this instance bootstrapping was used to prevent this. Later studies also employed this cut-off and found prognostic significance in their respective cohorts [14,25]. Several studies employed both cut-offs for HRR1 at 12 or 18 bpm, according to what exercise test was given to patients. Those who underwent standard exercise tests often employed a cut-off of 12 bpm, whereas echocardiographic exercise stress tests tended to employ an 18 bpm cut-off [9,26]. One explanation could be that echocardiographic tests did not employ an active recovery; instead, patients lay supine, which may account for differences in thresholds [27].

HRR2 is a second commonly employed time-point. Shetler et al. [9] first suggested that a threshold of 22 bpm was superior to other time-periods like HRR1. This finding was later supported by Lipinski et al. [28], who suggested that HRR2 was the best time-point at predicting survival, and Goda et al. [29], who found HRR2 to be a univariate predictor in their study.

Further studies have included the use of different thresholds according to statistical tests, such as ROC curve analyses. The study of Lipinski et al. [30] noted various cut-off points according to the presence of HF or left ventricular systolic dysfunction (LVSD) and whether the participants were taking beta-blockers (Table 2).

Table 2 represents the vast differences in thresholds employed at 1, 2, and 5 min and how factors like beta-blockers may affect thresholds. However, studies commonly use one cut-off value for their population based on previous larger cohort studies or determine their own, as there is no universally agreed upon time-point or threshold in the literature [14].

Sheppard et al. [19] calculated median HRR in their cohort and used this value to try and distinguish between survivors and non-survivors. They discovered that patients whose HRR was below the median (24 bpm) at 90 s were more likely to be hospitalised within the 5-year follow-up period but found no differences in mortality when using this median value at any time-point. The lack of significance found may be explained by the absence of pre-determined thresholds based on statistical tests.

Overall, there is no single universal method to obtain HRR and further investigation is required to ascertain the best method. However, the literature commonly employs HRR1 (12 bpm or 18 bpm) and HRR2 (22 bpm). Nevertheless, other thresholds and time-points have also shown prognostic significance.

### 3.3. Methodological Differences

#### 3.3.1. Exercise Mode

Treadmill and cycle ergometry were the two main exercise modes used across the literature. The decision to employ either a treadmill or a cycle ergometer appears to be dependent on an institution’s current practices. Although the use of a treadmill was more common, there appeared to be an increased use of cycle ergometers in the newer literature. Comparisons can be made between studies with different protocols. However, caution should be taken, as variables such as $\dot{V}$O_2_ peak, for example, at values which are 5–10% higher on a treadmill than cycle ergometer [31], as seen in Table 3.

The study by Watanabe et al. [16] of 5483 HF patients predominantly employed treadmill testing (93%). However, a small minority used bicycle testing, and no difference was found in the distribution of HRR according to exercise mode and no interaction between exercise mode and HRR in mortality prediction. Arena et al. [20] found that abnormal HRR1 could predict mortality when considering both exercise modalities: on a treadmill (hazard ratio (HR) 4.8, 95% CI 2.1–11.0, *p* < 0.001) and on a cycle ergometer (HR 7.0, 95% CI 3.8–13.2, *p* < 0.001). One novel study explored whether HRR would have prognostic value in a 6 min walk test (test of functional performance); the authors found that the predictive accuracy of HRR after the walking test was consistent with that of CPET in HF patients [32].

This would suggest that exercise mode had no significant effect on HRR or its prognostic capacity, though more studies are required to confirm these findings.

#### 3.3.2. Exercise Protocol

Three main protocols were used in the literature: Bruce protocol, step protocol, or individualised ramp protocol. Most of these tests resulted in a symptom-limited or maximal exercise; some studies included sub-maximal exercise tests.

The research of Tang et al. [33] included a retrospective sample of 202 HF patients; they all underwent graded increases in workload according to either the Bruce (n = 114) or the modified Naughton (n = 88) protocol. Post hoc analysis showed that the type of exercise protocol was significantly associated with mean HRR (Bruce 26 bpm vs. Naughton 20 bpm, *p* = 0.001). The Naughton protocol had a significantly slower HRR, which translated to increased risk of clinical outcomes (death and cardiac transplantation) in a univariate analysis (HR 5.8, 95% CI 1.7–19, *p* = 0.004). This study had a wide CI, potentially due to the smaller sample size or the relatively small number of events that occurred (n = 32). The researchers concluded that an attenuated HRR increased adverse outcomes in their study, noting that the choice of protocol or other unmeasured confounders may have influenced findings. Most studies did not individually comment on the protocol; however, the literature would suggest that HRR remains prognostically significant, independently of exercise protocol.

The focus across the literature has been on maximal exercise tests to measure HRR. Four studies included patients with sub-maximal exercise tests. Cole et al. [13] was the first study to postulate and consequently show that an attenuated HRR was predictive of mortality, independently of workload achieved, in those undergoing exercise testing. Bilsel et al. [25] found that, when exercise capacity was sub-maximal, an abnormal HRR (<18 bpm) was still a strong predictor of death. Yanagisawa et al.’s [14] study of octogenarians employed a cut-off in testing (target: 85% of predicted maximum heart rate), with only 64% of participants reaching this and many of the tests being sub-maximal in nature. Despite this, the authors concluded that abnormal HRR remained a significant prognostic marker. Cahalin et al.’s [34] research categorised patients undergoing CPET into subgroups according to peak respiratory exchange ratio (RER), which is a gauge of subject effort. A peak RER of ≥1.10 was considered to constitute maximal effort, and one below that threshold was considered to be a sub-maximal effort (RER < 1.10). They found that sub-maximal tests were equally predictive of mortality compared to maximal tests.

The literature suggests that the exercise protocol does not impact on the prognostic significance of HRR and sub-maximal exercise tests may also be a suitable protocol in the assessment of HRR. However, the studies used different definitions to describe sub-maximal tests (percent of maximum heart rate or RER), and so further research is needed to clarify these categorisations.

#### 3.3.3. Recovery

##### Active Cool Down

Early studies measured HRR using an active cool-down. The study of Cole et al. [13] was the first to explore the prognostic significance of HRR in 2428 patients who undertook CPET. Their protocol used an active cool-down period using a speed of 2.4 km/h at a grade of 2.5%; the study concluded that HRR1 was a powerful predictor of mortality. Nishime et al. [15] replicated this protocol and also validated previous findings. Cahalin et al. [34] also found prognostically significant results and suggested that the use of an active cool-down best reflects current clinical practice which increases HRR ecological validity.

##### Passive Cool-Down

Those studies, which measured HRR during a passive recovery, can be further subdivided into studies which placed patients in a supine position and those which asked them to stand or be seated. Shetler et al.’s [9] study of 2193 patients who had performed treadmill exercise testing was the first to measure the HRR of patients in the supine position. They found that HRR1 and HRR2 still had prognostic value, despite not having any cool-down period. Similarly, one study employed a passive recovery method of laying supine and also found HRR was predictive of mortality [28].

Bilsel et al.’s [25] study of 84 patients with HF performed symptom-limited exercise tests using a passive seated recovery. The authors also supported the prognostic significance of HRR1 using this protocol.

One study followed 2935 patients with suspected CAD. The majority of these (n = 2426) followed a recovery using a cool-down period. However, 509 patients did not have a cool-down period; the study concluded that abnormal HRR independently predicted mortality irrespective of the protocol used [27].

The majority of studies employed two different strategies: an active cool-down or immediate rest (supine or seated). There was no clear pattern in methodology regarding the cohort of patients. For example, disease states did not influence which protocol was performed. The literature would suggest that the presence or absence of a standardised cool-down does not impact the prognostic significance of HRR.

Overall, one distinct difference between the studies included in this literature review concerns the protocols employed. There has been little standardisation of exercise modality, protocol, or recovery period between studies; these differences are not based on the need to adapt to patient population. Thus, further research is required to validate an optimal protocol and to standardise testing for different clinical populations. Standardised trials will allow data to be pooled into single quantitative estimates of greater sample size and will enable the quality and strength of evidence to be assessed.

### 3.4. Prognosis

The importance of HRR is derived from its potential ability to predict increased morbidity and mortality within the assessed population; this would support the monitoring and risk assessments of patients.

#### 3.4.1. Heart Failure

##### All-Cause Mortality

Nissinen et al. [21] found abnormal HRR1 (<12 bpm) increased ACM in patients of NYHA class I-III (HR 5.3, 95% CI 1.5–18.5, *p* < 0.01). However, the most powerful predictor of ACM was HRR1 combined with low exercise capacity (<1.0 watt/Kg), (HR 9.8, 95% CI 3.5–27.8, *p* < 0.001), having 84% specificity and 70.6% sensitivity. In another study, 109 patients comprised the HF subgroup: abnormal HRR2 (<24 bpm) was a significant predictor of ACM in those not taking BB [30]. Nanas et al. [24] also provided evidence demonstrating that abnormal HRR1 (<12 bpm) was a strong predictor of mortality (OR 8.5, 95% CI 3.5–20.7, *p* < 0.001) in HF patients [30]. Moreover, they found that risk stratification was improved by HRR in patients who were categorised as ‘intermediate risk’ ($\dot{V}$O_2_ peak; 10–18 mL/kg/min) regarding $\dot{V}$O_2_ peak and $\dot{V}$_E_/$\dot{V}$CO_2_ (≤34).

One study did not support the prognostic significance of HRR1 (*p* = 0.08) in 390 patients with HF and a reduced ejection fraction; however, this study did not determine cut-off values for HRR which may have significantly affected the result [35]. Hajdusek et al. [23] failed to show the prognostic significance of HRR in the cohort of 78 patients with systolic HF. However, the sample size was small, and HRR was calculated as a slope of recovery, which has been previously shown to be a less effective means of measuring HRR [22].

##### Cardiovascular Mortality

Arena et al. [18] found that HRR1 outperformed $\dot{V}$O_2_ peak in predicting CVM or hospitalisations and provided additional prognostic information when assessed alongside the $\dot{V}$_E_/$\dot{V}$CO_2_ slope: abnormal $\dot{V}$_E_/$\dot{V}$CO_2_ (>34.4) and HRR1 (<6.5 bpm) increased risk (HR 9.2, 95% CI 4.5–18.5, *p* < 0.001) in comparison to the use of abnormal HRR1 alone (HR 4.6, 95% CI 2.3–9.3, *p* < 0.001). A later study found that the abnormal HRR1 threshold of ≥16 bpm was a significant marker for patients with both ischaemic HF (HR 3.3, 95% CI 1.8–6.0, *p* < 0.001) and non-ischaemic HF aetiology (HR 8.1, 95% CI 3.1–21.5, *p* < 0.001) and may outperform $\dot{V}$O_2_ peak in prognosis [20].

HRR1 in a univariate regression analysis was a significant predictor of CVM (HR 2.29, 95% CI 1.57–3.34, *p* < 0.001). A low HRR1 (<16 bpm), combined with an elevated $\dot{V}$_E_/$\dot{V}$CO_2_ (<34), is a strong predictor of mortality (HR 3.51, 95% CI 2.33–5.29, *p* < 0.001) in patients with an intermediate $\dot{V}$O_2_ (10.1–13.9 mL·kg/min) [36]. Furthermore, several other studies support the prognostic significance of HRR for CVM in HF patients [32,34]. Further studies support the prognostic significance of HRR, but used summed scores (i.e., HRR and $\dot{V}$O_2_ peak) to better improve predictive capacity [37]. The likelihood of survival over 2 years for patients displaying multiple risk factors is significantly lower than that for individuals with fewer risk factors (Figure 3). HRR was given a weighted score of 5. As such, abnormal HRR in addition to any other prognostic factor (e.g., $\dot{V}$_E_/$\dot{V}$CO_2_) decreased likelihood of survival [37].

The literature supports HRR as a good prognostic marker for both CVM and ACM endpoints in patients with HF.

#### 3.4.2. Coronary Artery Disease

##### All-Cause Mortality

Cole et al.’s [13] study had a 6-year follow-up with 213 deaths: HRR1 < 12 bpm strongly predicted death. This was inclusive of the 225 people who had CAD (Relative risk (RR) 3.2, 95% CI 1.6–6.6, *p* = 0.001). The authors showed HRR is a predictor of mortality with 56% sensitivity for low HRR (<12 bpm) and 77% specificity. In another study, with a median 5.2 year follow-up, HRR1 (<12 bpm) was again predictive of death in all groups; those with CAD (HR 2.6, 95% CI 1.86–3.78, *p* < 0.001) and without CAD (HR 4.25, 95% CI 3.19–5.66, *p* < 0.001) [15]. If HRR1 went below 10 bpm, survival at 5 years decreased significantly [15]. Patients with abnormal HRR1 (<12 bpm or <18 bpm according to exercise protocol) were at increased risk of death (17% vs. 9%, HR 2.0, 95% CI 1.8–2.3, *p* < 0.0001) in over 7000 patients who were prospectively followed for a mean of 3.7 years [26]. Additionally, sex was not shown to affect the prognostic significance of HRR, adding to the generalisability of HRR as a predictive measure. Yanagisawa et al.’s [14] study of 97 octogenarians: 61% had CAD and univariate Cox proportional-hazard analysis demonstrated abnormal HRR (≤18 bpm) was independently predictive of ACM (HR 2.82, 95% CI 1.06–7.47, *p* = 0.037). Again, abnormal HRR was prognostically significant for ACM [16]. Shetler et al. [9] found that low HRR was predictive of greater mortality, independent of the severity of CAD over a 7-year follow-up. This is supported by a study that showed in a univariate analysis that abnormal HRR at 1 min (<12 bpm) increased risk of mortality in a similar fashion with patients exhibiting any CAD (HR 1.9, 95% CI 1.5–2.4, *p* < 0.0001) and severe CAD (HR 2.0, 95% CI 1.6–2.6, *p* < 0.0001) [27]. Lipinski et al. [28] found HRR2 (<22 bpm) better discriminated survivors from non-survivors and predicted mortality (*p* < 0.001).

##### Cardiovascular Mortality

A study found that HRR was attenuated in non-survivors compared to survivors (*p* = 0.002) and found HRR2 (<22 bpm) to be a univariate predictor of CVM in their study (RR, 2.04, 95% CI 1.33 to 3.17, *p* = 0.001) [29]. Similarly, Karjalainen et al. [38] found HRR (<21 bpm) was predictive in univariate analyses (HR 1.6, 95% CI 1.1–2.2, *p* = 0.012).

The literature supports HRR as a good prognostic marker for both CVM and ACM in patients with CAD and HF; however, more research may be required to determine if HRR is better paired with other prognostic markers.

### 3.5. Pharmacological Impact

Beta-blockers (BB) are widely used to treat patients with CVD. BB operate by reducing HR, systolic BP, the risk of plaque rupture, and microvascular damage [39]. Notably, Beta-1 blockade (common in BB) is accountable for a 35% reduction in ACM [39]. As BB impact heart rate, their influence on the prognostic value of HRR in patients requires further investigation, especially due to previous disagreements in the literature [9,20].

Cole et al. [13] originally suggested that abnormal HRR was predictive of death, regardless of BB use. This finding has been supported by an author who found that abnormal HRR was predictive of mortality in the absence (HR 2.18, 95% CI 1.87–2.55, *p* < 0.0001) and presence (HR 1.65, 95% CI 1.26–2.18, *p* = 0.0004) of BB [26]. Similarly, no difference was found in survival curves between those on BB and not when considering an abnormal HRR [9].

Goda et al. [29] showed that BB use itself did not predict survival, with no difference being found between the number of survivors and non-survivors in 37.7% of participants (n = 550) who were taking BB. Additionally, Tang et al. [33] showed that the use of BB was not associated with HRR (BB 23 ± 13 vs. no-BB 24 ± 13 bpm) or clinical outcome. Multiple studies found minimal differences in HRR values between BB- and non-B-medicated patients [19,28], suggesting that HRR is mainly reflective of vagal tone and that HRR can still be used to risk-stratify all patients, irrespective of BB therapy [23].

There have been some disagreements between studies. Watanabe et al.’s [16] study found a weak interaction with lower mortality in patients with an abnormal HRR who were receiving BB. Conversely, Karjalainen et al. [38] observed an increase in short-term cardiovascular events with BB use (14% BB vs. 8% no BB). Differences in findings may be due to the relatively small sample of patients subdivided by BB use, which can limit the ability to draw conclusive findings. There was one study which asserted that HRR was not predictive of mortality in those taking BB [15]. This conclusion may have been made due to the small number of events that occurred in those taking BB.

In summary, the literature suggests that HRR is prognostically significant, regardless of BB use. However, further research should standardise a cut-off for abnormal HRR with BB use and examine whether BB use impacts survival or cardiovascular events when considering abnormal HRR.

## 4. Practical Applications

The findings of this review suggest that a slower HRR increases the risk of both CVM and ACM. Exercise interventions have looked at improving HRR through cardiac rehabilitation (CR) programmes and thus sought to improve prognosis.

Yaylalı et al. [40] found that a CR programme significantly improved HRR1 and HRR2 in patients with an abnormal HRR at baseline. Recent meta-analyses and reviews provided stronger evidence that CR can reduce cardiac mortality by ~26%, improve quality of life and reduce hospitalisations [41]. Future research should focus on finding the best method to improve fitness and prognostic variables, such as HRR, to improve outcomes in high-risk patients with CVD.

Practitioners should assess HRR in patients undergoing CPET with the aim of screening high-risk patients and improving prognosis through cardiac rehabilitation.

### Limitations

This paper’s validity is limited by publication bias, i.e., unfavourable findings are less likely to have been published and are likely not included in the review. There was also the risk of selection bias in studies included in the study. However, a search strategy with strict inclusion/exclusion criteria was established and implemented to reduce this risk.

## 5. Conclusions

The literature suggests that HRR measures provide valid prognostic information beyond that of typical exercise testing measures in both CAD and HF populations, independently of the exercise protocol or presence of BB. As such, this simple and non-invasive measure of autonomic function should be routinely incorporated into exercise test interpretation to aid in risk stratification and the prognosis of patients. Nevertheless, future research can use the literature summarised here to conduct studies with a view to developing a standardised methodology for using HRR as a prognostic variable in clinical populations with CVD.

## Figures and Tables

**Figure 1 ijerph-20-04678-f001:**
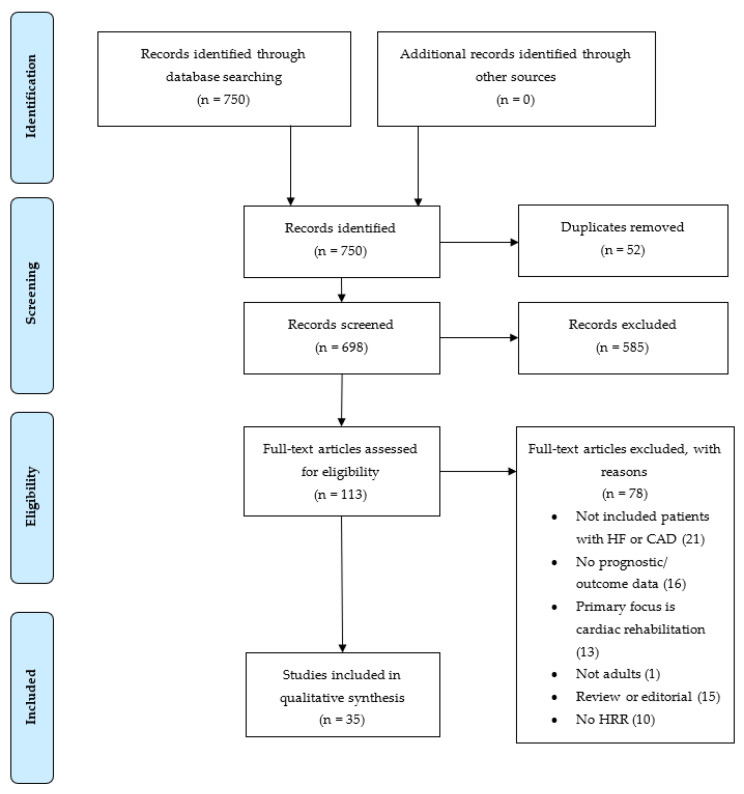
Study selection flow diagram showing number of records identified, screened, excluded, and included as part of the final review. HF = heart failure, CAD = coronary artery disease, HRR = heart rate recovery.

**Figure 2 ijerph-20-04678-f002:**
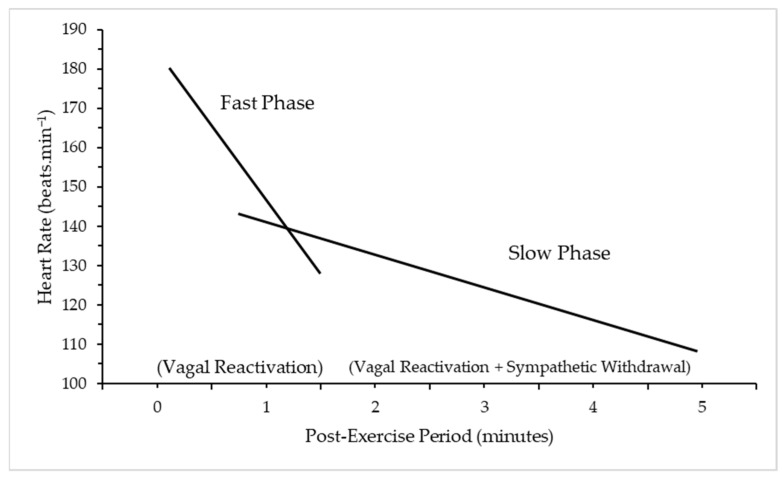
Fast and slow phase of Heart Rate Recovery (HRR) after exercise cessation. Adapted with permission from Peçanha, Silva-Júnior and Forjaz (2014) [17]. 2013, Wiley Publishers. A slow decrease in HRR represents an abnormality, with a sympathetic tone favoured as a result of the autonomic imbalance [18]. The pathology of this imbalance is unclear but a number of mechanisms have been proposed: changes in the SA node, or central control; down-regulation of cardiac beta-1 adrenoreceptors, which can decrease the range of sympathetic and parasympathetic activity and slow the response time [19]. The slower the HRR, the greater the risk of adverse outcomes [20].

**Figure 3 ijerph-20-04678-f003:**
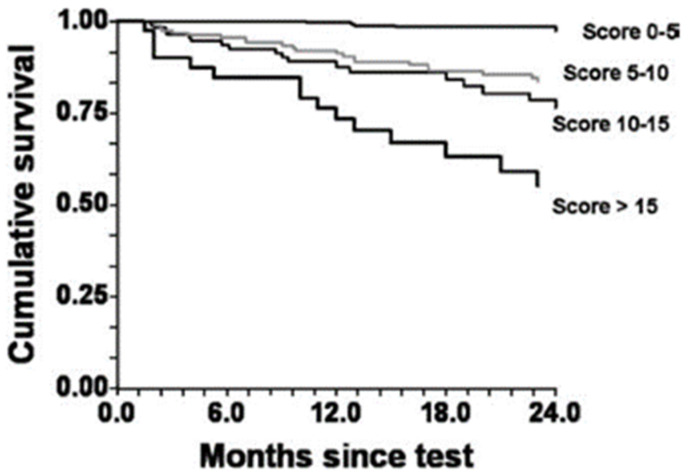
Cumulative survival by Kaplan–Meier analysis for composite risk scores. *p* < 0.001 by log-rank test. (Adapted with permission from Myers et al. [37]. 2008, Elsevier).

**Table 1 ijerph-20-04678-t001:** List of absolute and relative contraindications (Adapted with permission from [4]. 1997, American Heart Association).

Absolute	Relative
Acute myocardial infarctions (<2 days)	Left main coronary stenosis
Unstable angina	Moderate stenotic valvular disease
Uncontrolled cardiac arrhythmias causing symptoms of hemodynamic compromise	Electrolyte abnormalities
Symptomatic severe aortic stenosis	Severe arterial hypertension
Uncontrolled symptomatic heart failure	Tachyarrhythmias or bradyarrhythmias
Acute pulmonary embolus or pulmonary infection	Hypertrophic cardiomyopathy and other forms of outflow tract obstruction
Acute myocarditis and pericarditis	Mental or physical impairment, leading to inability to exercise adequately
Acute aortic dissection	High-degree atrioventricular block

**Table 2 ijerph-20-04678-t002:** Cut-point selection using ROC Curves for abnormal HRR at 1, 2, and 5 min after exercise in patients with LVSD, HF, or neither (Adapted with permission from [30]. 2005, Elsevier).

HRR	Cut-Point (bpm)	Predictive Accuracy	Sensitivity	Specificity
1 min HRR				
LVSD alone + no BB	Abnormal <9	61%	35%	78%
LVSD alone + BB	Abnormal <9	62%	44%	72%
HF + no BB	Abnormal <9	69%	65%	73%
HF+ BB	Abnormal <9	72%	64%	79%
Neither + no BB	Abnormal <10	63%	54%	66%
Neither + no BB	Abnormal <10	64%	52%	68%
2 min HRR				
LVSD alone + no BB	Abnormal <27	69%	48%	82%
LVSD alone + BB	Abnormal <23	70%	56%	78%
HF + no BB	Abnormal <24	64%	51%	78%
HF + BB	Abnormal <23	64%	45%	79%
Neither + no BB	Abnormal <24	71%	31%	86%
Neither + no BB	Abnormal <17	77%	25%	94%
5 min HRR				
LVSD alone + no BB	Abnormal <33	65%	30%	88%
LVSD alone + BB	Abnormal <30	69%	51%	79%
HF + no BB	Abnormal <37	65%	54%	78%
HF + BB	Abnormal <36	60%	55%	64%
Neither + no BB	Abnormal <32	72%	21%	90%
Neither + no BB	Abnormal <32	67%	43%	75%

LVSD = left ventricular systolic dysfunction, HF = heart failure, BB = beta blockers, HRR = heart rate recovery.

**Table 3 ijerph-20-04678-t003:** Cycle ergometry versus treadmill testing (Adapted with permission from [3]. 2003, American Thoracic Society.)

	Cycle Ergometer	Treadmill Ergometer
Maximal oxygen uptake	Lower	Higher
Work rate measurement	Yes	No
Blood gas collection	Easier	More difficult
Noise and artifacts	Less	More
Safety	Safer	Less safe
Weight bearing in obese	Less	More
Degree of leg muscle training	Less	More
More appropriate for	Patients	Active healthy subjects

## Data Availability

Not applicable.

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
