# Peer review of "Heart Rate Recovery Assessed by Cardiopulmonary Exercise Testing in Patients with Cardiovascular Disease: Relationship with Prognosis"

_ijerph, 2023, doi:10.3390/ijerph20064678_

Round 1

Reviewer 1 Report

Congratulations for your very well-written review-article that indeed covers an scare area whose importance and frequency of recommendation is continuously growing. I do not have modifications to make to your article apart from the limitations you mentioned, except the suggestion to add some ideas concerning the use of cardiopulmonary testing (precisely, HRR) to predict the need for transplantation. Can we use this parameter in addition to the already known VO2max and VE/VCO2 that have already proven to be predictions of mortality if low in the population of HF subjects on the transplantation lists?

Author Response

Reviewer 1 Comments:

Congratulations for your very well-written review-article that indeed covers an scare area whose importance and frequency of recommendation is continuously growing. I do not have modifications to make to your article apart from the limitations you mentioned, except the suggestion to add some ideas concerning the use of cardiopulmonary testing (precisely, HRR) to predict the need for transplantation. Can we use this parameter in addition to the already known VO2max and VE/VCO2 that have already proven to be predictions of mortality if low in the population of HF subjects on the transplantation lists?

Response to Reviewer: Thank you for your response. Regarding the use of CPET data to predict the need for cardiac transplantation there is a paucity of data; even using traditional CPET variables such as VO2 max, much less for heart rate recovery. With more recent advancements in survival/outcomes for heart transplant candidates, those who are considered more stable patients are often not listed for heart transplants. Typically, heart transplant candidates have recurrent hospitalisations and are unable to perform CPET. Thus, the ability to perform exercise testing characterises a ‘low-risk’ population who are then excluded from transplantation listing (1). However, there are some studies which have shown that VE/VCO2 slope can be used in assessing patients risk and as a predictor of survival benefit in heart transplant patients (2).  Interestingly, Giardini et al., (3) provided some initial support for HRR use as a prognostic tool in paediatric heart transplant patients and demonstrated that HRR represented a valuable measurement which may help to inform medical management. However, more evidence is required to gain a better understanding of heart rate recovery as a prognostic marker in heart failure patients undergoing transplantation and at present is beyond the scope of this review.

References:

  1. Corrà U, Agostoni PG, Anker SD, Coats AJS, Leiro MGC, Boer RA De, et al. Role of cardiopulmonary exercise testing in clinical stratification in heart failure . A position paper from the Committee on Exercise Physiology and Training of the Heart Failure Association of the European Society of Cardiology. Eur J Heart Fail. 2018;20:3–15.
  2. Ferreira A, Tabet J-Y, Frankenstein L, Metra M, Mendes M, Zugck C, et al. Ventilatory Efficiency and the Selection of Patients for Heart Transplantation. Circ Hear Fail. 2010;3:378–86.
  3. Giardini A, Fenton M, Derrick G, Burch M. Impairment of heart rate recovery after peak exercise predicts poor outcome after pediatric heart transplantation. Circulation. 2013;128(SUPPL.1):199–204.

Reviewer 2 Report

The paper presents the Heart Rate Recovery Assessed by Cardiopulmonary Exercise Testing in Patients with Cardiovascular Disease: Relationship with Prognosis . Overall, the scientific objective is important and interesting. The article is well written and comprehensive. There are some  revisions needed. Please provide a point-by-point response to the following queries.

1.       please explain the abbreviations used in the figures and tables in their footnotes.

2.       Was the review done according to PRISMA Guidelines?

3.       Please provide a flowchart, which depicts the flow of information through the different phases of a systematic review. It should include detailed information on the number of records identified, included and excluded, and the reasons for exclusions.

4.       The methodology section should provide information on full electronic search, including any limits used, such that it could be repeated. Specify study characteristics and report characteristics (e.g., years considered, language, publication status) used as criteria for eligibility, giving rationale.

5.       The results should information on numbers of studies screened, assessed for eligibility, and included in the review, with reasons for exclusions at each stage.

 6.  Figure 1 and 2 are not acceptable, since they are published already in journals, are protected by copyright. Please create your own graphs to replace the copyrighted figures.

Author Response

Heart Rate Recovery Assessed by Cardiopulmonary Exercise Testing in Patients with Cardiovascular Disease: Relationship with Prognosis

Firstly, we would like to thank the reviewers for taking the time to review our manuscript. We appreciate the comments and have considered them, please see our responses to each below.

Reviewer 2 Comments:

The paper presents the Heart Rate Recovery Assessed by Cardiopulmonary Exercise Testing in Patients with Cardiovascular Disease: Relationship with Prognosis . Overall, the scientific objective is important and interesting. The article is well written and comprehensive. There are some revisions needed. Please provide a point-by-point response to the following queries.

  1. please explain the abbreviations used in the figures and tables in their footnotes.

Response to Reviewer: Thank you for your comment, abbreviations have been explained throughout the manuscript now.

  1. Was the review done according to PRISMA Guidelines?

Response to Reviewer: Thank you for this comment. No, the review was initially performed with the intention of writing a narrative review, however in order to ensure the review was performed with rigor, we framed the review in the style of a systematic narrative review hence the use of specific search terms, across a set time period with relevant inclusion and exclusion criteria.

  1. Please provide a flowchart, which depicts the flow of information through the different phases of a systematic review. It should include detailed information on the number of records identified, included and excluded, and the reasons for exclusions.

Response to Reviewer: A helpful comment, thank you. A study selection flow chart has been included in the manuscript now under Figure 1 to show this information.

  1. The methodology section should provide information on full electronic search, including any limits used, such that it could be repeated. Specify study characteristics and report characteristics (e.g., years considered, language, publication status) used as criteria for eligibility, giving rationale.

Response to Reviewer: Thank you for this suggestion, we have amended the manuscript in the methods section under ‘Data Sources’ and ‘Study Selection’ to highlight the inclusion and exclusion criteria for records in our review as per your suggestion. We have also amended the search terms used to source articles for added clarity in the methods section.

  1. The results should information on numbers of studies screened, assessed for eligibility, and included in the review, with reasons for exclusions at each stage.

Response to Reviewer: Thank you for this comment. As above, a study selection flow chart has been included in the manuscript now under Figure 1 to show this information.

  1. Figure 1 and 2 are not acceptable, since they are published already in journals, are protected by copyright. Please create your own graphs to replace the copyrighted figures.

Response to Reviewer: Thank you for this comment. We have now adapted figure 2 with permission from the original author and noted this in the caption of figure 2. The new figure has been created ourself.

We have obtained permission from the original author to include the new figure 3. A comment has been added in the manuscript to show permission was obtained from the original author in the caption for figure 3. An official letter has been included as a separate document from the author as proof.

Round 2

Reviewer 2 Report

Thank you for the replies and the changes to the text. I have no further remarks.